# CT-Navigated Spinal Instrumentations–Three-Dimensional Evaluation of Screw Placement Accuracy in Relation to a Screw Trajectory Plan

**DOI:** 10.3390/medicina58091200

**Published:** 2022-09-01

**Authors:** Arthur Gubian, Lisa Kausch, Jan-Oliver Neumann, Karl Kiening, Basem Ishak, Klaus Maier-Hein, Andreas Unterberg, Moritz Scherer

**Affiliations:** 1Department of Neurosurgery, Heidelberg University Hospital, 69120 Heidelberg, Germany; 2Division of Medical Image Computing, German Cancer Research Center (DKFZ) Heidelberg, 69120 Heidelberg, Germany; 3Pattern Analysis and Learning Group, Department of Radiation Oncology, Heidelberg University Hospital, 69120 Heidelberg, Germany

**Keywords:** pedicle screw accuracy, three-dimensional accuracy, spinal navigation, spinal instrumentation, navigation-guided spine surgery, Gertzbein–Robbins classification

## Abstract

*Background and Objectives:* In the literature, spinal navigation and robot-assisted surgery improved screw placement accuracy, but the majority of studies only qualitatively report on screw positioning within the vertebra. We sought to evaluate screw placement accuracy in relation to a preoperative trajectory plan by three-dimensional quantification to elucidate technical benefits of navigation for lumbar pedicle screws. *Materials and Methods*: In 27 CT-navigated instrumentations for degenerative disease, a dedicated intraoperative 3D-trajectory plan was created for all screws. Final screw positions were defined on postoperative CT. Trajectory plans and final screw positions were co-registered and quantitatively compared computing minimal absolute differences (MAD) of screw head and tip points (mm) and screw axis (degree) in 3D-space, respectively. Differences were evaluated with consideration of the navigation target registration error. Clinical acceptability of screws was evaluated using the Gertzbein–Robbins (GR) classification. *Results*: Data included 140 screws covering levels L1-S1. While screw placement was clinically acceptable in all cases (GR grade A and B in 112 (80%) and 28 (20%) cases, respectively), implanted screws showed considerable deviation compared to the trajectory plan: Mean axis deviation was 6.3° ± 3.6°, screw head and tip points showed mean MAD of 5.2 ± 2.4 mm and 5.5 ± 2.7 mm, respectively. Deviations significantly exceeded the mean navigation registration error of 0.87 ± 0.22 mm (*p* < 0.001). *Conclusions*: Screw placement was clinically acceptable in all screws after navigated placement but nevertheless, considerable deviation in implanted screws was noted compared to the initial trajectory plan. Our data provides a 3D-quantitative benchmark for screw accuracy achievable by CT-navigation in routine spine surgery and suggests a framework for objective comparison of screw outcome after navigated or robot-assisted procedures. Factors contributing to screw deviations should be considered to assure optimal surgical results when applying navigation for spinal instrumentation.

## 1. Introduction

Spinal pedicle screw fixation is a surgical technique widely used for treatment of a large range of pathologies (spinal trauma, tumors, deformity, degenerative conditions) [1]. Because of its proximity to critical structures (spinal cord, dural sac, nerve roots and blood vessels), screw placement accuracy is essential for the success of the procedure, with potential complications ranging from dural leak to neurological complications including paralysis [2,3]. Free-hand fluoroscopic technique carries a risk of screw misplacement, with rates of 20 to 40% in the thoracolumbar spine [4,5]. This led to the development and the introduction of navigation systems, which helped to greatly reduce the rate of screw misplacement to reported misplacement rates as low as 0.3% [6]. In addition to navigation systems, several robotic navigation-based assistance systems were recently developed with the goal of improving the screw placement accuracy and surgical efficacy.

A number of prospective as well as retrospective works were published to demonstrate the accuracy of navigated or robotic screw placement. In the literature, screw accuracy has predominantly been evaluated according to the screw position within the vertebra on postoperative CT as noted in a meta-analysis performed by Aoude et al. [7]. Several screw placement grading systems have been described (e.g., Gertzbein and Robbins (GR) being the most widely used [8], but also Wiesner [9] and Rampersaud [10]), which generally assess the degree of violation of the pedicular cortex based on a 2-D evaluation on standard planar reconstructions.

The literature abounds on comparisons of navigated or robotic screw placement to free-hand placement under fluoroscopy but only a few studies focused on direct evaluation of screw results after robotic-assisted or navigated placement with regard to screw trajectory plans [2,11,12,13]. The primary question that arises in this context nowadays is not only the achieved screw position within the pedicle but the evaluation of achieved screw accuracy with regard to the previously planned screw trajectory in 3D-space [12].

While GR grade and other 2-dimensional assessments were sufficient to show how the used guidance technology helped to improve surgical results in comparison to free-hand techniques, the specific 3-dimensional evaluation of how reliable a trajectory plan can be transferred to the surgical site is required for accurate comparison of navigated and robotic results in order to fully elucidate the technical benefits of these technologies in spine surgery.

In this study, we compared final screw positions of CT-navigated lumbar and sacral pedicle screws with corresponding intraoperative screw trajectory plans and performed 3D-quantitative assessment of placement accuracy to address this issue for navigated spinal instrumentations.

## 2. Materials and Methods

### 2.1. Patients and Study Design

We performed a retrospective evaluation of prospectively collected data. From a consecutive institutional registry of CT-navigated spinal instrumentations with dedicated screw planning performed at the authors’ institution between January 2010 and December 2018 (*n* = 1660), we randomly selected *n* = 30 cases with lumbosacral surgery performed by a single surgeon for this analysis. All cases in this study had intraoperative CT (NavCT) for navigated screw insertion as well as postoperative CT (ControlCT) to validate screw positions.

All screw plans included the following parameters: screw head point defining the cortical entry of the screw into the vertebra, screw tip point locating the screw tip within the vertebral body and screw length and diameter as chosen by the surgeon.

For evaluation of possible contributors to screw accuracy, clinical data were extracted from patient records.

The institutional review board approved the processing of anonymized data for evaluation of screw accuracy in spinal instrumentation and the requirement for informed consent was waived (S-723/2017). Figure 1 gives a flow-chart of case selection and an overview of image analysis.

### 2.2. Surgical Procedure for CT-Navigated Instrumentations

All procedures were performed as standard open approaches by a single senior spine surgeon with over 20 years of experience in complex spine surgery under general anesthesia in prone position. Standard polyaxial titanium alloy screw systems (Expedium^®^, Depuy Synthes Company, Raynham, MA, USA; XIA^®^, Stryker Corporation, Kalamazoo, MI, USA) were used, adapting length and diameter to the patient’s individual anatomy. All navigation procedures were performed based on intraoperative CT (Siemens, CT Emotion^®^, Sliding Gantry, Siemens Company, Erlangen, Germany) covering the desired spinal levels. Patient registration was performed using “point-to-point navigation” as described previously [6]. For this registration method, a minimum of three *reference screws* (4 mm long, 1.5 mm thick, Zimmer Biomet Holdings, Inc., Warsaw, IN, USA. usually used for cranial plating) are placed on the laminae prior to intraoperative CT. *Reference screws* are then used for co-registration by serving as unique anatomical landmarks. Mean Navigation registration error was calculated by the navigation system as a target registration error in millimeter (mm) and was recorded for analysis.

Intraoperative screw trajectory plans were created on the navigation workstation by the operating surgeon based on intraoperative CT data (Stryker Navigation System Nav3i with SpineMap™ 3D-Navigation, Stryker Corporation, Kalamazoo, MI, USA).

Screw trajectories were planned as standard lumbar pedicle screws following the convergent pedicular anatomy and with parallel orientation to the superior endplate of the vertebra. Individualized screw trajectories following bone mineral density (e.g., cortical bone screws) were not investigated in this study.

This created virtual 3D screw illustrations of screw positions including trajectory, length and diameter. For screw insertion, the virtual illustrations were used as drill paths to be followed with tracked surgical instruments. The pedicle tract was prepared using a navigated awl. After insertion of a K-Wire in the created tract, the screw was inserted.

As a general principle, the operating surgeon in charge for cases analyzed in this study maintained cautious adherence to virtual drill paths whenever possible. The acceptance of planned drill paths for screw implantation during surgery was assured according to the surgical records for all cases in this study. Fluoroscopy was only used after screw implantation for final verification of screw positioning.

### 2.3. Postoperative Image Analysis

For each case, intraoperative navCT contained the information of the screw trajectory plan (screw dimensions and respective trajectories), as defined by the surgeon. Postoperative controlCT was retrieved for each case to confirm final screw positions. On each controlCT, final screw positions were manually segmented using the Stryker SpineMap 3D Software, by tracing the depicted screw with the screw planning tool. This defined each final screw position as a volumetric object on controlCT. Segmentation of final screw position was performed in consensus with two spine surgeons blinded to corresponding intraoperative screw plans.

For evaluation of final screw position in relation to the intraoperative trajectory plan, navCT and corresponding controlCT were rigidly co-registered using the MultiModal.rigid-Algorithm integrated in the Medical Imaging Interaction Toolkit (MITK v2021.02, MITK.org, Heidelberg, Baden-Wurttemberg, Germany). Appropriate co-registration was verified according to the boundaries of the vertebral bodies of the covered spinal levels. Verification was performed collaboratively by two authors specialized in spine surgery.

Subsequently, intraoperative screw plans and final screw positions were imported as volumetric objects linked to their respective source CT to MITK. This generated a superimposition of screw plans with corresponding final screw positions. Under consideration of co-registration of the respective source CTs, this allowed for an accurate planned versus actual comparison of screw positioning within 3D space (See Figure 1 for an overview of image analysis and Figure 2 and Figure 3 for illustration of analyzed screw parameters).

### 2.4. D-Quantitative Evaluation of Screw Positions

For 3D-quantitative evaluation, the minimal absolute distances (MAD) of planned and actual head and tip points (in millimeters) as well as the angular deviation of screw direction (in degrees) were automatically calculated for each screw by customizing Python scripts from the NumPy Package (v1.20). An illustration of the evaluated parameters and points is given in Figure 2. The linear measurements (deviation between planned and final screw head and tip positions and angular deviation) were calculated as Euclidian distances (distance between two points in 3D space (x, y, z coordinates and their differences d)) according to the following formula (Formula 1):(1)$$MAD=\sqrt{d{\left(x\right)}^{2}+d{\left(y\right)}^{2}+d{\left(z\right)}^{2}}$$

Screw length and diameters were evaluated by comparing their respective mean values. Results were evaluated under consideration of the mean navigation registration error obtained after registration of imaging to the patient during spinal navigation.

### 2.5. Qualitative Evaluation of Screw Positions Using Gertzbein–Robbins

For qualitative evaluation, all final screw positions on respective controlCT were reviewed and scored according to the Gertzbein–Robbins (GR)-Classification in consultation with two independent spine surgeons blinded to trajectory plans and 3D-quantitative results. Screws were classified under this system according to the following: grade A, no cortical breach; grade B, 0 to <2 mm of cortical breach; grade C, 2 to <4 mm of cortical breach; grade D, 4 to <6 mm of cortical breach; and grade E, ≥6 mm of cortical breach [8]. Screws graded as either A or B were qualitatively classified as clinically acceptable as it is handled in the existing literature [8,12,14]. All screws were planned to correspond to GR grade A on intraoperative screw plans.

### 2.6. Statistics

All continuous variables from quantitative screw evaluation were evaluated by their means and standard deviation. Demographic patient data was evaluated by median and interquartile range (IQR). One-way ANOVA with Holm–Sidak’s post-test for multiple comparisons was used for evaluation of intergroup variances in quantitative analysis and evaluation of MADs in spinal levels. Intergroup differences of quantitative screw measures and qualitative GR grades were evaluated by the *t*-test. Distributions of qualitative measures across spinal levels addressed was performed with the X2-Test. Demographic patient data was evaluated by the non-parametric Kruskal–Wallis test. *p*-values < 0.05 were regarded as statistically significant. Data composition was performed using Excel (Microsoft Corp., Redmond, WA, USA) and the statistical analysis was performed with Graph Pad Prism 9 (GraphPad Software, San Diego, CA, USA)

## 3. Results

### 3.1. Population

From a consecutive registry of CT-navigated instrumentations, 30 cases from a single surgeon were randomly selected for analysis. Surgery reports confirmed intraoperative acceptance of planned screw trajectories for all cases selected. Upon final intraoperative fluoroscopy after screw implantation, all screw positions were acceptable and no screw was revised after navigated implantation.

After co-registration of navCT with corresponding controlCT, three cases had to be excluded from the analysis because of an inaccurate co-registration affecting the cranial spinal segments on visual inspection. The inaccuracies observed precluded precise evaluation of planned screw trajectories with final screw positions in these cases.

The final cohort used for analysis of screw accuracy consisted of 27 cases with lumbosacral instrumentations. Our cohort covered spinal segments L1-S1 and evaluated accuracy of *n* = 140 screws in total (L1: *n* = 2, L2: *n* = 8, L3: *n* = 24, L4: *n* = 40, L5: *n* = 44, S1: *n* = 22). Median construct length was two segments (six screws) ranging from one segment (four screws) to four segments (ten screws). Among the included cases, the median age was 69.0 (IQR 61–72) years and the median BMI was 29.5 kg/m^2^ (IQR 25.3–32.2). All cases underwent open procedures and the indication for surgery was degenerative pathology for 25 cases (including spondylolisthesis grade 1 and 2) and infectious pathology for 2 cases (pyogenic spondylodiscitis). The demographic characteristics of included cases are presented in Table 1.

### 3.2. D-Quantitative Evaluation of Screw Positions

After co-registration of navCT and control-CT and superimposition of corresponding screw plans, quantitative evaluation of final screw positions in relation to the intraoperative screw plan was performed using a Python script. For the intraoperative screw plan and final screw position a planned versus actual analysis was performed using the deviation of screw axis (in degree) and MAD for corresponding screw head and tip points (in mm) in 3D space, respectively (Table 2).

For screw direction, implanted screws showed a mean deviation of 6.3 ± 3.6° (range 0.7–16.1°) compared to the trajectory plan. For screw head points, MAD between screw plan and final position was 5.2 ± 2.4 mm (range 0.5–12.9 mm). For screw tip points, MAD was 5.5 ± 2.7 mm (range 1.2–18.3 mm) (Table 2 and Figure 4). These differences were evaluated on the basis of a mean navigation registration error of 0.87 ± 0.22 mm (range 0.5–1.2 mm). Differences observed in the planned vs. actual comparisons were significantly greater compared to the mean navigation registration error (*p* < 0.001) for screw direction, head and tip points, respectively (Figure 4).

To evaluate the influence of the spinal level addressed on screw accuracy, the quantitative analysis was extended to cover each spinal level separately (Figure 5). The L1 level was excluded from statistical analysis due to limited screw data (*n* = 2 screws). For screw direction, we observed comparable deviations among spinal levels L2-L5 (4.8 ± 1.5°; 5.9 ± 4.0°; 5.3 ± 2.8°; 6.4 ± 3.6°, respectively), while deviations in direction were significantly greater at S1 (9.3 ± 4.0°, *p* < 0.001). For screw head points, we found no significant differences among spinal levels L2-S1 (4.8 ± 1.8; 5.3 ± 2.0; 4.8 ± 2.1; 4.9 ± 2.6; 6.3 ± 2.6 mm, *p* = 0.1, respectively). For screw tip points, planned vs. actual deviations were comparable at L2-L5 (4.2 ± 1.5; 5.0 ± 2.1; 4.7 ± 1.8; 5.5 ± 2.6 mm, respectively) but significantly increased at S1 (7.5 ± 3.7, *p* < 0.001) This illustrated increased degrees of freedom for screw instrumentation a S1 compared to other spinal levels, respectively.

### 3.3. Evaluation of Screw Positions Using Gertzbein–Robbins

All screws were additionally evaluated according to the Gertzbein–Robbins classification. In 129 screws (92.1%), positioning was scored GR grade A and 11 screws (7.9%) were scored GR grade B. No final screw position was scored grade C or worse in this study (Table 1). All final screw positions evaluated in this study were clinically acceptable corresponding to either grade A or B according to GR. We found no statistically significant association between the GR-grade and spinal segment addressed in this cohort (*p* = 0.23) (Figure 6). We additionally evaluated the interplay between quantitative and GR screw outcomes in this study. When analyzing 3D-quantitative measures in GR grade A vs. grade B screws, we did not find statistically significant differences for either direction (*p* = 0.24), head point (*p* = 0.20) or tip point (*p* = 0.19). Nor did we observe a significant difference in navigation registration error in GR grade A compared to grade B screws (0.87 ± 0.23 mm vs. 0.89 ± 0.26 mm, *p* = 0.75), respectively.

We evaluated possible correlations of demographic factors and GR screw outcome in univariate regression or contingency analysis. We observed a statistical trend for higher age in cases affected by GR grade B compared to grade A screws (median 70y (IQR 69–78) vs. 69y (IQR 60–72), *p* = 0.056). There was a trend for higher weight in GR grade B compared to grade A screws (median 90 kg (IQR 77–95) vs. 83 kg (IQR 76–95), *p* = 0.052). However, BMI was no significant confounder for qualitative screw outcome in our study. Additionally, treated spinal pathology had no impact on screw outcome. See Table 2 for details.

## 4. Discussion

In this study, we sought to assess the accuracy obtained in navigation-guided placement of lumbar and sacral pedicle screws by 3D-comparison to a screw trajectory plan. The deviation between planned screws and final screw position on postoperative CT was calculated automatically in 3D space using a Python script. While screw placement was clinically acceptable in all 140 screws according to GR (100%), considerable deviation in implanted screws compared to the initial trajectory plan was noted in 3D-quantitative analysis exceeding the navigation’s registration error.

In this discussion, we provide a critical appraisal of factors contributing to screw deviation known from the literature and put them into context of our findings.

### 4.1. Evaluation of Screw Accuracy

A review of recent literature on screw placement accuracy in navigated lumbosacral instrumentation based on intraoperative imaging was performed since 2010. Table 3 summarizes findings from original articles and meta-analyses and illustrates that the majority of studies focused on categorical grading systems (GR) or binary classifications (i.e., the presence of cortical breach) to investigate screw outcome. According to GR, our findings are in line with the literature achieving clinically acceptable screw positions in 90–100% of cases in navigated procedures [2,11,14,15,16,17,18,19,20], which is clearly superior to freehand or fluoroscopy-guided placement of pedicle screws ranging from 57.0% to 98.3% [8,21,22].

While GR and similar grading proved suitable to underscore the contribution of navigation to improved screw accuracy and to establish navigated screw placement as a routine procedure in contemporary spine surgery, these grading systems reach their limits when it comes to evaluation of different navigation setups or to elucidating benefits of robotic-assisted over navigated procedures, however. In order to thoroughly evaluate technical benefits of these techniques, 3D-quantitative evaluation of screw accuracy in terms of a planned vs. actual comparison is required.

This motivated our quantitative analysis which indicated that despite all screws being rated clinically acceptable, considerable variability between planned and actual screw position was observed. Only a few studies in the literature quantified screw accuracy regarding preoperative plans or according to anatomical landmarks [2,12,13]. Miller et al. have compared planned trajectories to final screw positions after navigation in this regard, but their evaluation was limited to measuring mean angular deviations to the midsagittal line (2.22°) and superior endplate (2.61°) in 2D axial and sagittal views, respectively [2]. A recent analysis from Jiang et al. presented an in-depth analysis of robotic screw placement accuracy based on a pre-planned trajectory using 3D quantifications in 254 screws. In their evaluation, clinical screw outcome was comparable to our results showing clinical acceptability in all screws analyzed (184 (72%) GR grade A, 70 (28%) GR grade B screws, respectively). In their 3D-quantitative analysis for lumbosacral screws (*n* = 191), the authors report mean deviations of 3.8 ± 2.4 mm for screw tip, 4.2 ± 2.5 mm for screw head and 3.8 ± 2.8° for angular screw deviation, respectively [12]. Given our analysis of 3D-accuracy in navigation, these findings suggest superior results in a planned vs. actual comparison for robotic-assisted placed screws using a similar methodology, which could point out a benefit for robot assistance in instrumentation procedures.

It has to be noted, however, that Jiang et al. relied on manual measurements of screw deviations involving a risk for measurement bias in their analyses. In contrast, our proposed approach brings the advantage of using automatic, coding-script based calculation of Euclidian distances between the relevant screw parameters according to point coordinates in 3D space, yielding an objective quantification of screw accuracy.

### 4.2. Technical Aspects of Screw Deviations

The navigation target registration error defines the degree of precision that is obtained after patient-to-image registration during surgery. This was found to average 0.87 ± 0.22 mm (range 0.5–1.2 mm) in our study, joining the existing literature on the subject (0.74 to 0.75 mm [12]). Additionally, there is a possibility of incorrect registration of surgical tools (up to 2.3 mm during bench testing according to the manufacturer) [2]. In our analysis, we acknowledged the registration error as a systematic error occurring during navigated screw implantation. Our statistical analysis indicated that observed deviations of screw head and tip points exceeded the registration error, suggesting additional sources to be contributing to the planned vs. actual deviation observed.

Deflection of the navigation reference array or intersegmental shifts of the spine alignment during the instrumentation in patients with spine instability could negatively impact on navigation accuracy in multi-segment constructs [2]. As an advantage of the point-to-point navigation used in our study, accuracy of navigation can be checked prior to each screw using the reference screws serving as unique anatomical landmarks and revision of patient-to-image registration can be performed at all times, if required. Moreover, the point-to-point registration method used has been shown to be robust to anatomical and patient-related variations in navigation accuracy in our experience [6].

Patient-related factors or compromised imaging quality were not relevant contributors to screw deviation from the initial trajectory plan in our study. Specifically, obesity, although being a suggestive risk factor for screw misplacement in the literature, was not associated with increased quantitative or qualitative deviation in our analysis (Table 2) [14].

Eventually, one has to be aware of the risk for the aforementioned technical errors to conjoin during the procedure, leading to an amplification of initially small deviations to yield a larger final amount [14].

### 4.3. Clinical Importance of Screw Deviations

As we illustrated by our GR evaluation, screw outcome was clinically acceptable in all implanted screws. This satisfactory result puts the specific clinical relevance of our quantitative analysis into question. There are multiple approaches to elucidate “minimal clinically important differences” (MCID) in spine surgery, which essentially discusses the impact of measured statistical differences on the health of a patient [34]. Regarding the lumbar spine, Miller et al. proposed a MCID for screw angulation to be ±5° from an ideal trajectory until critical cortical breach at the pedicle side is expected [2]. Hence, the 6° mean angular deflection observed in our cohort illustrates how CT-navigation provides high reliability in respecting a proposed 10° corridor for acceptable screw placement in lumbosacral surgery.

By 3D-quantitative analysis we were also able to demonstrate possible differences in accuracy to robotic procedures, which has not been done previously. While our screws were placed within 5 mm of the planned position with a mean deflection of 6°, Jiang et al. reported greater accuracy by robotic implantation with 4 mm and 4° overall deviation, respectively [12].

While this difference will unlikely have a current impact with regard to MCID, future aspects of spinal instrumentation have to be considered to perceive the potential significance: computer assisted screw planning with subsequent guided instrumentation has the potential to improve screw stability and to reduce screw-associated complications by optimization of screw positioning under consideration of local bone quality, corresponding fastening strength and other anatomical constraints [35,36,37]. The knowledge about and quantification of limitations on accuracy of navigated or robotic systems is essential to improve current technology, which will have an impact on outcome with the introduction of new features to spine surgery in the future.

An aspect of clinical relevance in navigated surgery were significantly greater screw variances at the sacral compared to all lumbar levels [2]. While entrance points in our cohort correlated well with the initial plan, head position and screw trajectory differed significantly from the initial plan at the sacrum. In our opinion, this is a result of anatomical challenges for angulation and tissue retraction and also reflects the greater degrees of anatomical freedom at the sacrum compared to the lumbar levels, where the pedicle anatomy mostly defines the screw trajectory. This points out another advantage for robotic systems, which have been reported with comparable performance for thoracic, lumbar and sacral levels [12]. 3D-quantification of thoracic screws should be pursued to extend accuracy analysis for CT-navigated screws.

### 4.4. Limitations

The monocentric and retrospective nature of our study entails a potential for selection bias limiting the generalizability of our results. Our approach of random sampling of cases from a consecutive cohort reflects our efforts to identify a representative sample of routine surgical cases in an academic spine center. The limitation to a single surgeon was chosen since a focused adherence to planned screw paths during surgery was required for evaluation of navigated screw accuracy. This prerequisite was provided given the surgeon’s individual technique for instrumentation and adherence to planned screw trajectories was checked in surgery reports, additionally.

Since all screws were planned by a single surgeon, our findings can only carefully be generalized to other surgeons and setups. Even though the planned vs. actual comparison performed in our study should apply similarly for other surgeons, the influence of individual preferences for screw placement, alternative screw trajectories (e.g., cortical bone screws) and the influence of different navigation systems on accuracy has not been investigated in our study. Our series overlooks *n* = 140 screw positions covering spinal levels L1-S1 from an institution with extensive experience in CT-based spinal navigation. While lumbar pedicles usually are of round diameter and are aligned with the superior endplates, thoracic pedicle anatomy can be significantly narrower and with steep inclination angles, which has possible implications for screw placement and accuracy in a planned vs. actual comparison. Moreover, the impact of severe spinal pathology e.g., high-grade spondylolisthesis or spinal deformities was not specifically evaluated in our study. Hence, reported accuracy cannot generally be transferred to the thoracic spine or to other spinal pathologies treated.

The accuracy of the co-registration of intraoperative navCT with postoperative controlCT impacted on the quality of our quantitative measurements. Problems with rigid co-registration can originate from spinal alignment changes and segmental shifts induced during the respective procedures. In order to minimize this bias, appropriate co-registration was visually verified by two independent spinal surgeons blinded to trajectory plans. Cases with non-satisfactory co-registrations (*n* = 3) were excluded from the analysis due to shifts affecting the cranially located vertebrae.

There is a potential for measurement bias when screw positions or angular deviations are compared by manual measurements, as it was described by several studies performing quantitative comparisons [2,7,12,14]. A strength of our approach was that all screw comparisons were performed automatically by directly comparing point coordinates of respective screw parameters within 3D space using a Python script. This made any manual measurements unnecessary as angular deviations and MAD of screw parameters were calculated automatically. This analytic framework enables accurate and objective comparisons of screw plans with final screw positions.

## 5. Conclusions

Screw placement was clinically acceptable in all screws evaluated in this analysis but nevertheless, considerable deviation in implanted screws was noted compared to the initial trajectory plan. Our data provides a 3D-quantitative benchmark for screw accuracy achievable by CT-navigation in routine spine surgery and suggests a framework for future objective comparison of screw outcome after navigated and robot-assisted procedures. Factors contributing to screw deviations should be considered to assure optimal surgical results when applying navigation for spinal instrumentation.

## Figures and Tables

**Figure 1 medicina-58-01200-f001:**
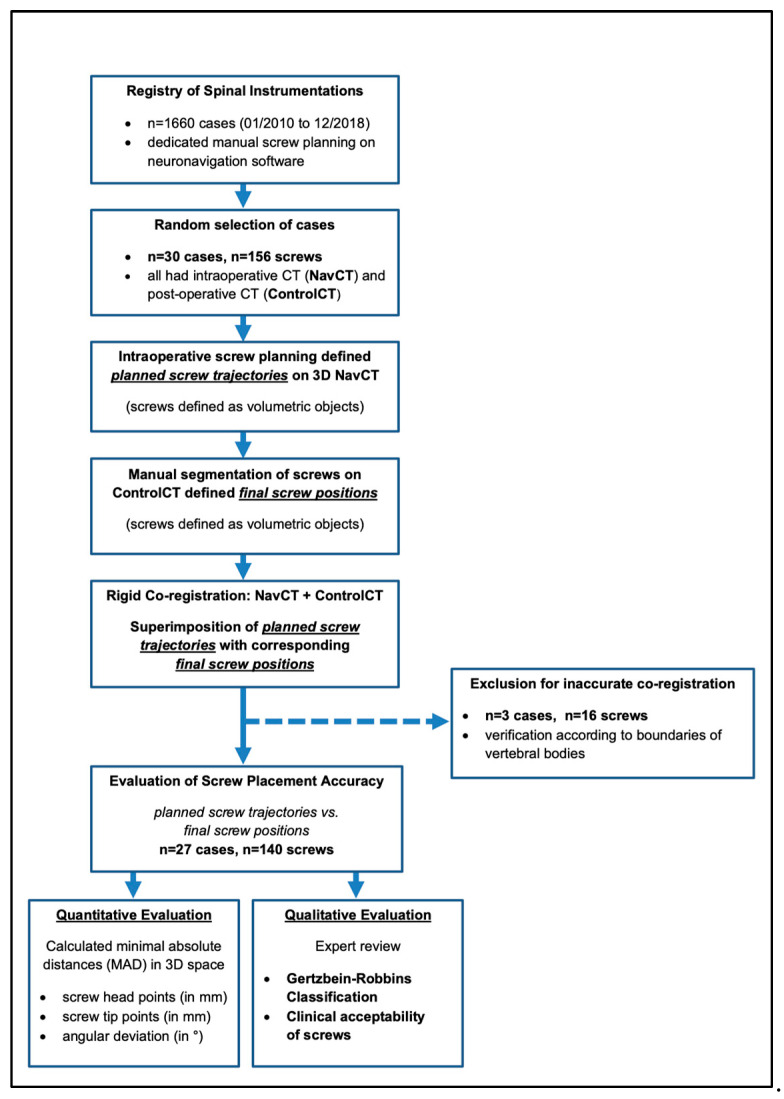
Flow diagram of case selection and workflow of data analysis.

**Figure 2 medicina-58-01200-f002:**
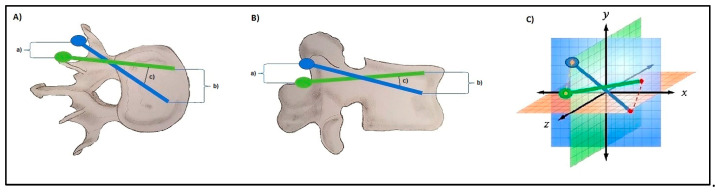
Graphic illustration of the methodology for measuring screw accuracy: Euclidian distances were calculated in 3D-space from point coordinates to yield mean absolute distances (MAD) between screw heads (a), screw tips (b) and angular deviation (c) between planned screw trajectory (exemplified in green) and final screw position (exemplified in blue); Methodology illustrated for axial (**A**) and sagittal (**B**) planes and in 3D-space (**C**).

**Figure 3 medicina-58-01200-f003:**
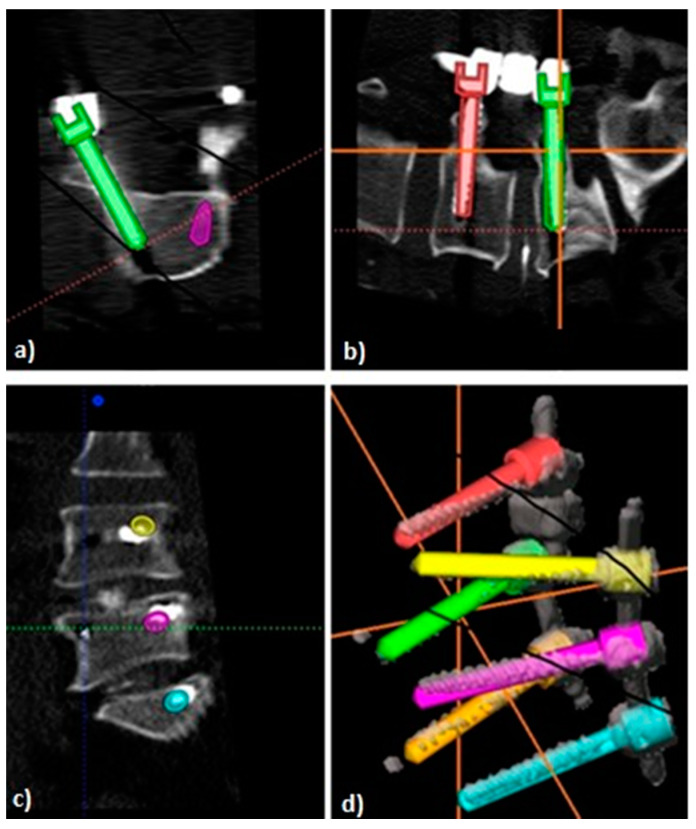
Illustration of superimposed screw positions for analysis of screw placement accuracy. Planned screw trajectories (colored screw illustrations) are superimposed on postoperative control-CT depicting final screw positions (white screw illustrations). Illustrations for transverse (**a**), sagittal (**b**), coronal (**c**) views and for 3D-reconstructions (**d**).

**Figure 4 medicina-58-01200-f004:**
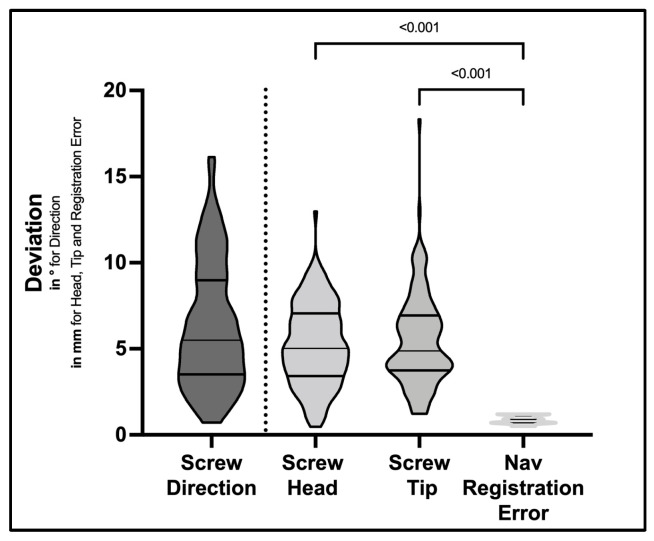
Deviation of final screw positions to preplanned screw trajectories illustrated in violin plots. Results are presented as angular deviation (in degrees) between screws for screw direction and mean absolute distances (in mm) between screw head and screw tip points. Differences for head and tip points were statistically significant in relation to navigation registration error recorded during surgery, respectively (ANOVA *p* < 0.001).

**Figure 5 medicina-58-01200-f005:**
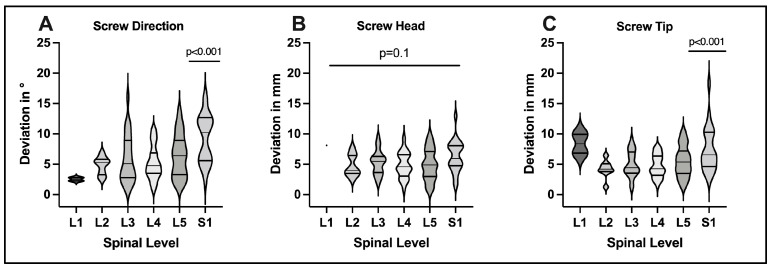
Evaluation of screw accuracy for screw direction (**A**), screw head (**B**) and screw tip (**C**) in relation to the spinal level treated. While deviation of final screw position and preplanned trajectory were comparable across spinal levels for screw head (**B**) (*p* = 0.1), significantly increased deviations were observed at the S1 level for screw direction (**A**) (*p* < 0.001) and screw tip (**C**) (*p* < 0.001), respectively. L1 was excluded from this statistical analysis due to paucity of data (*n* = 2 screws).

**Figure 6 medicina-58-01200-f006:**
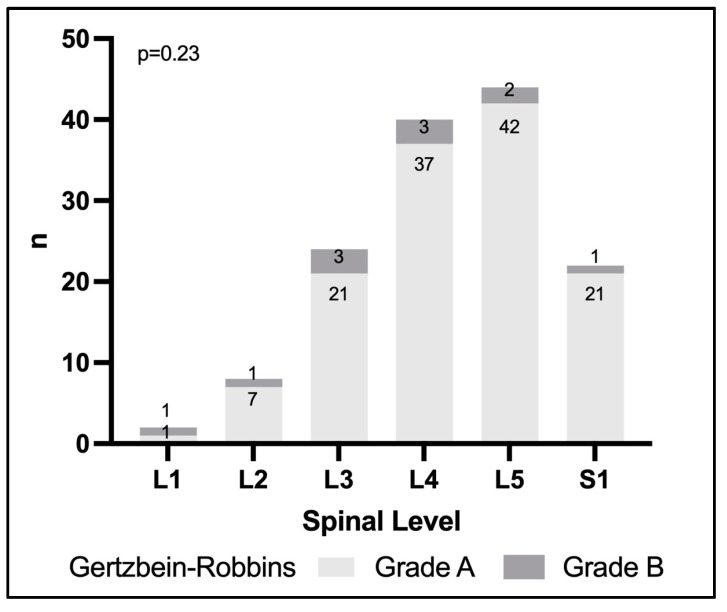
Qualitative evaluation of final screw position according to the Gertzbein–Robbins classification across spinal levels treated. There were no differences in qualitative screw results across spinal levels (*p* = 0.23).

**Table 1 medicina-58-01200-t001:** Demographics.

*n* **(Cases)**		**27**
**Age** (years, median, IQR)		69.0 (61.0–72.0)
**Weight** (Kg, median, IQR)		85.0 (76.0–95.0)
**Size** (cm, median, IQR)		170.0 (158.0–181.0)
**BMI** (Kg/m^2^, median, IQR)		29.5 (25.3–32.3)
**Diagnosis** *n* (%)	**Degenerative Spondylolisthesis°1**	15 (55.5)
	**Degenerative Spondylolisthesis°2**	1 (3.7)
	**Spondylosis/Stenosis**	9 (33.3)
	**Spondylodiscitis**	2 (7.4)
**ASA score** *n* (%)	**2**	15 (55.5)
	**3**	11 (40.7)
	**NA**	1 (3.8)
**Nicotine abuse** *n* (%)	**no**	10 (37.0)
	**yes**	17 (63.0)
**Diabetes mellitus** *n* (%)	**no**	7 (25.9)
	**yes**	20 (74.1)
**Construct Length in** **Spinal Segments** *n* (%)	**1**	12 (44.4)
	**2**	10 (37.0)
	**3**	3 (11.1)
	**4**	2 (7.5)
**Screws Evaluated** **per Spinal Segment** *n* (%)	**L1**	2 (1.4)
	**L2**	8 (5.7)
	**L3**	24 (17.1)
	**L4**	40 (28.6)
	**L5**	44 (31.4)
	**S1**	22 (15.8)
**Gertzbein–Robbins** **Classification** *n* (%)	**A**	129 (92.1)
	**B**	11 (7.9)
	**≥C**	0 (0.0)

BMI: Body mass index, ASA: American Society of Anesthesiologists score.

**Table 2 medicina-58-01200-t002:** Three-Dimensional Quantitative Analysis of Screw Accuracy and Confounders of Qualitative Screw Accuracy.

		Grouped by Gertzbein–Robbins Classification			
	Overall	A	B	*p*-Value	Test
*n*	140	129	11		
**MAD Screw Direction** **(degrees, mean ± SD)**	6.3 ± 3.6	6.2 ± 3.5	7.5 ± 4.6	0.24	*t*-Test
**MAD Head Point** **(mm, mean ± SD)**	5.2 ± 2.4	5.1 ± 2.3	6.1 ± 3.0	0.20	*t*-Test
**MAD Tip Point** **(mm, mean ± SD)**	5.5 ± 2.7	5.4 ± 2.6	6.5 ± 2.8	0.19	*t*-Test
**Navigation** **registration error** **(mm, mean ± SD)**	0.87 ± 0.22	0.87 ± 0.23	0.89 ± 0.26	0.75	*t*-Test
**Age** **(Years, median, IQR)**	69.0 (61.0–72.0)	69.0 (60.0–72.0)	70.0 (69.0–78.0)	0.056	Kruskal–Wallis
**BMI** **(Kg/m^2^, median, IQR)**	29.5 (25.3–32.3)	29.4 (26.0–32.3)	30.4 (25.0–31.5)	0.914	Kruskal–Wallis
**Weight,** **(Kg, median, IQR)**	85.0 (76.0–95.0)	83.0 (76.0–95.0)	90.0 (77.0–95.0)	0.052	Kruskal–Wallis
**Diagnosis (*n*, %)**					
**Degenerative** **Spondylolisthesis°1**	70 (50.0)	64 (49.6)	6 (54.5)	0.776	X^2^-Test
**Degenerative** **Spondylolisthesis°2**	4 (2.9)	4 (3.1)			
**Spondylosis/** **Stenosis**	50 (35.7)	47 (36.4)	3 (27.3)		
**Spondylodiscitis**	16 (11.4)	14 (10.9)	2 (18.2)		

MAD: Minimal absolute differences, BMI: Body mass index, ASA: American Society of Anesthesiologists, SD: Standard deviation, IQR: Interquartile range.

**Table 3 medicina-58-01200-t003:** Review of the recent literature on pedicle screw accuracy in navigated lumbosacral instrumentation based on intraoperative imaging.

(A) Original Articles						
First Author (Year)	Imaging System	Navigation System	Number of Patients (*n* Screws)	Spine Region	Grading Method	Results
Scheufler (2011)	intraoperative CT	Brainlab	55 (826)	T/L/S	Heary, Gertzbein–Robbins Python	98.4% Heary I–III, 1.6% Heary IV 94.4% GR A, 4.6% GR B, 1% GR C
de Kelft (2012)	O-arm	Stealth Station	353 (1922)	T/L/S	Qualitative: correct/misplaced	95.7% of screws correctly placed
Waschke (2013)	intraoperative CT	Stealth Station	505 (2422)	T/L/S	Modified Gertzbein–Robbins Python	83.3% GR A, 11.2% GR B, 4.5% GR C, 1% GR D
Miller (2016)	O-arm	Stealth Station	31 (240)	T/L/S	Quantitative: screw trajectory vs. midsagittal line and superior endplate	2.17° ± 2.20° axial deviation 2.16° ± 2.24° sagittal deviation
Cordemans (2017)	ConeBeam CT	Brainlab	118 (695)	T/L/S	Heary, Gertzbein–Robbins Python	Breach rate 11.7% GR, 15.4% Heary
Laudato (2018)	O-arm	O arm navigation unit	25 (191)	T/L/S	Rampersaud criteria	69.6% grade A, 4.2% Grade B
Hecht (2018)	intraoperative CT 3D C-Arm	Brainlab	260 (1453)	C/T/L/S	Gertzbein–Robbins Python	69% GR A, 25% GR B, 5% GR C, 1% GR D
Ishak (2019)	intraoperative CT	Stryker	(1384)	C/T/L/S	Gertzbein–Robbins Python	92.6% GR A, 5.8% GR B, 1.4% GR C, 0.4% GR D, 0.7% GR E
Crawford (2020)	O-arm	O-arm navigation	84 (454)	C/T/L/S	pedicle breach yes/no	pedicle breach in 55 (12.1%)
Kumar (2020)	O-arm	Stealth Station S7	219 (1152)	C/T/L/S	Gertzbein–Robbins Python	overall breach rate: 3.6% 2.29% GR B, 0.82% GR C, 96.4% accuracy of pedicle screw fixation
Ouchida (2020)	O-arm	Stealth Station S7	138 (763)	T/L/S	3 points qualitative grading system	9.7% of screws deviated at postop CT evaluation
Sundaram (2020)	O-arm	Stealth Station S7	(2240)	T/L/S	Gertzbein–Robbins Python	accuracy 97.5%; overall breach rate 2.59%; major breach rate 0.94%, screw revision rate 0.7%
Kendlbacher (2022)	intraoperative CT, coneBeam CT, O-arm	Brainlab, Stealth Station S7	503 (2673)	C/T/L/S	Gertzbein–Robbins Python	92.1% GR A + B, 7.9% GR C + D
**(B) Meta-Analyses/Literature Reviews**						
**First Author** **(Year)**	**Grading Method**			**Goal of the Article/Title**		**Central Finding**
Tian (2011)	Pedicle screw violation (qualitative)			Pedicle screw insertion accuracy with different assisted methods: a systematic review and meta-analysis of comparative studies		pedicle breach less in CT navigated screws than in the conventional group (OR 95% CI 0.32–0.60)
Gelalis (2011)	Pedicle screw violation (qualitative, GR classification)			Accuracy of pedicle screw placement: a systematic review of prospective in vivo studies comparing free hand, fluoroscopy guidance and navigation techniques		violation in free hand technique range from 31 to 6%, vs. 11 to 0% with CT Navigation and 19 to 8% with fluoroscopy navigation
Liu (2017)	Qualitative			Accuracy of pedicle screw placement based on preoperative computed tomography versus intraoperative data set acquisition for spinal navigation system		higher accuracy in O-arm assisted screw insertion than in CT based navigation method (RR1.96; 95% CI 1.05–3.64; *p* = 0.03)
Du (2018)	3 categories: (1) excellent (screw completely within pedicle), (2) acceptable (a portion of the screw outside the pedicle 1 mm), and (3) poor (a portion of the screw outside the pedicle > 1 mm)			Accuracy of pedicle screw insertion among 3 image-guided navigation systems: systematic review and meta-analysis		prevalence of pedicle violation lower in the 3D fluoroscopy navigation than the 2D fluoroscopy navigation and the CT navigation
Perdomo-Pantoja (2019)	Qualitative: cortical breach: minor (<4 mm) or major (4 mm)			Accuracy of current techniques for placement of pedicle screws in the spine: a comprehensive systematic review and meta-analysis of 51,161 screws		CT Navigation showed the highest pedicle screw placement accuracy (95.5%)
Feng (2020)	Qualitative			O-arm navigation versus C-arm guidance for pedicle screw placement in spine surgery: a systematic review and meta-analysis		accuracy with O-arm navigation higher than with C-arm fluoroscopy.

A summarizes original articles [2,6,11,14,16,17,19,20,23,24,25,26,27]. Literature reviews and meta-analyses are represented in B [28,29,30,31,32,33] C = cervical, T = thoracic, L = lumbar, S = sacral, GR = Gertzbein–Robbins PythonClassification: Grade A: no breach, B < 2 mm breach, C > 2–4 mm breach, D > 4–6 mm breach, E > 6 mm breach.

## Data Availability

The data that support the findings of this study are available on request from the corresponding author. The data are not publicly available due to contained information compromising privacy necessitating informed consent.

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
