# Peer review of "CT-Navigated Spinal Instrumentations–Three-Dimensional Evaluation of Screw Placement Accuracy in Relation to a Screw Trajectory Plan"

_medicina, 2022, doi:10.3390/medicina58091200_

Round 1

Reviewer 1 Report

In this article, the authors describe a very interesting three-dimensional quantitative measurement method to evaluate screw placement accuracy in relation to the preoperative trajectory plan. They claimed that considerable deviation in implanted screws was noted compared to the initial trajectory plan. The paper is very well written and has certain enlightenment significance for the improvement of surgical navigation technique for lumbar degenerative diseases. The author's research provided a valuable tool for future objective comparison of screw positions after lumbar surgical procedures. The relevant research results are expected to improve the placement accuracy of pedicle screws after clinical application and have certain clinical application value in the future.

There are some problems, which must be solved before it is considered for publication. If the following problems are well-addressed, the reviewer believe that the essential contribution of this paper are important for the improvement of surgical navigation for lumbar pedicle screws.

1. The diameter and inclination angle of pedicles may affect the accuracy of screw placement. The relevant parameters and analysis of pedicle in selected cases are not mentioned in the article, whichshould be discussed. 

2. Bone mineral density may also affect the placement of pedicle screws, and the relevant parameters were not mentioned in the study.

3. The screw placement in the study was performed by a single doctor, which may cause research bias, and this point should be discussed in the article.

Author Response

We would like to thank the reviewers for their invaluable comments and remarks. We have
made extensions to our manuscript and hope to have addressed all remarks adequately.
Below you find a point-by-point reply to all comments. All replies are highlighted in italics.

Thank you for considering our manuscript for publication.

Sincerely,

Moritz Scherer MD, PhD

Reviewer #1

In this article, the authors describe a very interesting three-dimensional quantitative
measurement method to evaluate screw placement accuracy in relation to the preoperative
trajectory plan. They claimed that considerable deviation in implanted screws was noted
compared to the initial trajectory plan. The paper is very well written and has certain
enlightenment significance for the improvement of surgical navigation technique for lumbar
degenerative diseases. The author's research provided a valuable tool for future objective
comparison of screw positions after lumbar surgical procedures. The relevant research results
are expected to improve the placement accuracy of pedicle screws after clinical application
and have certain clinical application value in the future.

There are some problems, which must be solved before it is considered for publication. If
the following problems are well-addressed, the reviewer believe that the essential contribution
of this paper are important for the improvement of surgical navigation for lumbar pedicle
screws.

1. The diameter and inclination angle of pedicles may affect the accuracy of screw
placement. The relevant parameters and analysis of pedicle in selected cases are not
mentioned in the article, which should be discussed.

Thank you for this comment discussing the influence of pedicle anatomy on our results. We
have added a paragraph to the discussion section mentioning the influence of pedicle anatomy,
specifically in thoracic vs. lumbar spinal segments and mentioned this as a limitation to our
results in the updated version of the manuscript. (ll.431-438@p.15)

2. Bone mineral density may also affect the placement of pedicle screws, and the
relevant parameters were not mentioned in the study.

Screws planned and evaluated in our study were standard lumbar pedicle screws following the
convergent pedicular anatomy and with parallel orientation to the superior endplate of the
vertebra. Individualized screw trajectories following bone mineral density (e.g. cortical bone
screws) were not investigated in this study. We have added this detailed information to the
methods section for clarification (ll.116-119@p4) and mentioned this issue in the discussion
section as a limitation (ll.424-429@pp.14/15) in the updated version of the manuscript.

3. The screw placement in the study was performed by a single doctor, which may
cause research bias, and this point should be discussed in the article.

Thank you for this remark and indeed, all screws were performed by a single doctor. This had
advantages in ensuring the diligent adherence to the screw plans during screw implantation
but limits the generalizability of our results to other surgeons and setups. We have now
extended our discussion mentioning this as a limitation (ll.424-429@p.14/15)

Reviewer #2

First, I would like to thank the Authors for the quality of their work, with special attention to
the discussion. I do not have major comments, but only a few minor comments/suggestions.

Summary
Based on a retrospective evaluation of prospectively collected data, the Authors compared
CT-navigated lumbar and sacral pedicle screws' effective position with corresponding
intraoperative trajectory plans. 3 parameters (screw head point, screw tip point within the
vertebral body, angular deviation) and a qualitative assessment (Gertzbein-Robbins
classification and clinical acceptability of screws) were obtained. Although all screws were
clinically well placed, the Authors underlined a significant deviation compared to the planned
trajectory, exceeding the navigation’s registration error provided. An interesting discussion of
the main hypothesis follows this main result, with a quick review of the literature.

Strengths
Automatic measurements using home-built scripts combined with an open-source
software (MITK) for CT acquisitions co-registration. May the scripts be shared for
potential upcoming works from other teams?
Thank you for addressing this issue helping to share information regarding our methods with
other groups. All calculations of distances were performed using open source software.
Image analysis and co-registration was performed using MITK (v2021.02). Calculations of
minimal absolute distances (MAD) were performed using Python by customizing scripts from
the NumPy Package (v1.20) (numpy.org). We have added this information to our manuscript
at ll.139-141@p.4 and l.165@p.5, respectively.
Quick review of the literature with a comparison of recent works (Table 3).

Minor comments

1. L99@P3: Surgeon experience?
Thank you for pointing out this lack of detail in our methods section. We have now added
details to the surgeon’s experience, who was a senior spine surgeon with over 20 years
of experience in complex spine surgery. (l.100@p.3)

2. L135@P4: MITK is open-source software with multiple possible declinations.
Which version and modules were used?
Thank you for this remark requesting more detail on our image processing methods. We
used the MITK v2021.02 for screw evaluation and used the integrated MuliModal.rigid
Algorithm for co-registration of our images. We have added these details to the
manuscript now. (ll139-141@p.4)

3. L192@P6: OR is not explicitly defined

We have adjusted this to “surgery reports” instead of OR-records for clarification in the
whole manuscript now.

4. L354@P13: typo (pints instead of points)
That has been corrected, thank you. (l.360@p.13)

Figures and Tables Quality
Figures and Tables: ok

Ethics
Approved by the Institutional Review Board

Reviewer 2 Report

First, I would like to thank the Authors for the quality of their work, with special attention to  the discussion. I do not have major comments, but only a few minor comments/suggestions.

Summary

Based on a retrospective evaluation of prospectively collected data, the Authors compared CT-navigated lumbar and sacral pedicle screws' effective position with corresponding intraoperative trajectory plans. 3 parameters (screw head point, screw tip point within the vertebral body, angular deviation) and a qualitative assessment (Gertzbein-Robbins classification and clinical acceptability of screws) were obtained. Although all screws were clinically well placed, the Authors underlined a significant deviation compared to the planned trajectory, exceeding the navigation’s registration error provided. An interesting discussion of the main hypothesis follows this main result, with a quick review of the literature.

Strengths

- Automatic measurements using home-built scripts combined with an open-source software (MITK) for CT acquisitions co-registration. May the scripts be shared for potential upcoming works from other teams?

- Quick review of the literature with a comparison of recent works (Table 3).  

Minor comments

-          L99@P3: Surgeon experience?

-          L135@P4: MITK is open-source software with multiple possible declinations. Which version and modules were used?

-          L192@P6: OR is not explicitly defined

-          L354@P13: typo (pints instead of points

Figures and Tables Quality

-          Figures and Tables: ok

Ethics

-          Approved by the Institutional Review Board

Author Response

We would like to thank the reviewers for their invaluable comments and remarks. We have
made extensions to our manuscript and hope to have addressed all remarks adequately.
Below you find a point-by-point reply to all comments. All replies are highlighted in italics.

Thank you for considering our manuscript for publication.

Sincerely,

Moritz Scherer MD, PhD

Reviewer #1

In this article, the authors describe a very interesting three-dimensional quantitative
measurement method to evaluate screw placement accuracy in relation to the preoperative
trajectory plan. They claimed that considerable deviation in implanted screws was noted
compared to the initial trajectory plan. The paper is very well written and has certain
enlightenment significance for the improvement of surgical navigation technique for lumbar
degenerative diseases. The author's research provided a valuable tool for future objective
comparison of screw positions after lumbar surgical procedures. The relevant research results
are expected to improve the placement accuracy of pedicle screws after clinical application
and have certain clinical application value in the future.

There are some problems, which must be solved before it is considered for publication. If
the following problems are well-addressed, the reviewer believe that the essential contribution
of this paper are important for the improvement of surgical navigation for lumbar pedicle
screws.

1. The diameter and inclination angle of pedicles may affect the accuracy of screw
placement. The relevant parameters and analysis of pedicle in selected cases are not
mentioned in the article, which should be discussed.

Thank you for this comment discussing the influence of pedicle anatomy on our results. We
have added a paragraph to the discussion section mentioning the influence of pedicle anatomy,
specifically in thoracic vs. lumbar spinal segments and mentioned this as a limitation to our
results in the updated version of the manuscript. (ll.431-438@p.15)

2. Bone mineral density may also affect the placement of pedicle screws, and the
relevant parameters were not mentioned in the study.

Screws planned and evaluated in our study were standard lumbar pedicle screws following the
convergent pedicular anatomy and with parallel orientation to the superior endplate of the
vertebra. Individualized screw trajectories following bone mineral density (e.g. cortical bone
screws) were not investigated in this study. We have added this detailed information to the
methods section for clarification (ll.116-119@p4) and mentioned this issue in the discussion
section as a limitation (ll.424-429@pp.14/15) in the updated version of the manuscript.

3. The screw placement in the study was performed by a single doctor, which may
cause research bias, and this point should be discussed in the article.

Thank you for this remark and indeed, all screws were performed by a single doctor. This had
advantages in ensuring the diligent adherence to the screw plans during screw implantation
but limits the generalizability of our results to other surgeons and setups. We have now
extended our discussion mentioning this as a limitation (ll.424-429@p.14/15)

Reviewer #2

First, I would like to thank the Authors for the quality of their work, with special attention to
the discussion. I do not have major comments, but only a few minor comments/suggestions.

Summary
Based on a retrospective evaluation of prospectively collected data, the Authors compared
CT-navigated lumbar and sacral pedicle screws' effective position with corresponding
intraoperative trajectory plans. 3 parameters (screw head point, screw tip point within the
vertebral body, angular deviation) and a qualitative assessment (Gertzbein-Robbins
classification and clinical acceptability of screws) were obtained. Although all screws were
clinically well placed, the Authors underlined a significant deviation compared to the planned
trajectory, exceeding the navigation’s registration error provided. An interesting discussion of
the main hypothesis follows this main result, with a quick review of the literature.

Strengths
Automatic measurements using home-built scripts combined with an open-source
software (MITK) for CT acquisitions co-registration. May the scripts be shared for
potential upcoming works from other teams?
Thank you for addressing this issue helping to share information regarding our methods with
other groups. All calculations of distances were performed using open source software.
Image analysis and co-registration was performed using MITK (v2021.02). Calculations of
minimal absolute distances (MAD) were performed using Python by customizing scripts from
the NumPy Package (v1.20) (numpy.org). We have added this information to our manuscript
at ll.139-141@p.4 and l.165@p.5, respectively.
Quick review of the literature with a comparison of recent works (Table 3).

Minor comments

1. L99@P3: Surgeon experience?
Thank you for pointing out this lack of detail in our methods section. We have now added
details to the surgeon’s experience, who was a senior spine surgeon with over 20 years
of experience in complex spine surgery. (l.100@p.3)

2. L135@P4: MITK is open-source software with multiple possible declinations.
Which version and modules were used?
Thank you for this remark requesting more detail on our image processing methods. We
used the MITK v2021.02 for screw evaluation and used the integrated MuliModal.rigid
Algorithm for co-registration of our images. We have added these details to the
manuscript now. (ll139-141@p.4)

3. L192@P6: OR is not explicitly defined

We have adjusted this to “surgery reports” instead of OR-records for clarification in the
whole manuscript now.

4. L354@P13: typo (pints instead of points)
That has been corrected, thank you. (l.360@p.13)

Figures and Tables Quality
Figures and Tables: ok

Ethics
Approved by the Institutional Review Board.